# Insights into Growth Factors in Liver Carcinogenesis and Regeneration: An Ongoing Debate on Minimizing Cancer Recurrence after Liver Resection

**DOI:** 10.3390/biomedicines9091158

**Published:** 2021-09-04

**Authors:** Ana I. Álvarez-Mercado, Albert Caballeria-Casals, Carlos Rojano-Alfonso, Jesús Chávez-Reyes, Marc Micó-Carnero, Alfredo Sanchez-Gonzalez, Araní Casillas-Ramírez, Jordi Gracia-Sancho, Carmen Peralta

**Affiliations:** 1Department of Biochemistry and Molecular Biology II, Faculty of Pharmacy, University of Granada, 18071 Granada, Spain; 2Institute of Nutrition and Food Technology, Biomedical Research Center, University of Granada, 18016 Armilla, Spain; 3Instituto de Investigación Biosanitaria ibs.GRANADA, Complejo Hospitalario Universitario de Granada, 18014 Granada, Spain; 4Hepatic Ischemia-Reperfusion Injury Department, Institut de Recerca Biomèdica August Pi i Sunyer (IDIBAPS), 08036 Barcelona, Spain; acabalca31@alumnes.ub.edu (A.C.-C.); rojano@clinic.cat (C.R.-A.); mico@clinic.cat (M.M.-C.); 5Facultad de Medicina e Ingeniería en Sistemas Computacionales Matamoros, Universidad Autónoma de Tamaulipas, Matamoros 87300, Mexico; jesus.chavez@uat.edu.mx (J.C.-R.); acasillas@docentes.uat.edu.mx (A.C.-R.); 6Teaching and Research Department, Hospital Regional de Alta Especialidad de Ciudad Victoria “Bicentenario 2010”, Ciudad Victoria 87087, Mexico; asg_4@live.com; 7Liver Vascular Biology Research Group, Barcelona Hepatic Hemodynamic Laboratory, IDIBAPS Biomedical Research Institute, CIBEREHD, 03036 Barcelona, Spain; Jordi.gracia@idibaps.org; 8Barcelona Hepatic Hemodynamic Laboratory, Centro de Investigación Biomédica en Red de Enfermedades Hepáticas y Digestivas (CIBERehd), 08036 Barcelona, Spain

**Keywords:** liver cancer, liver resection, growth factors, regeneration, hepatocyte growth factor, insulin-like growth factor-1, epidermal growth factor

## Abstract

Hepatocellular carcinoma has become a leading cause of cancer-associated mortality throughout the world, and is of great concern. Currently used chemotherapeutic drugs in the treatment of hepatocellular carcinoma lead to severe side effects, thus underscoring the need for further research to develop novel and safer therapies. Liver resection in cancer patients is routinely performed. After partial resection, liver regeneration is a perfectly calibrated response apparently sensed by the body’s required liver function. This process hinges on the effect of several growth factors, among other molecules. However, dysregulation of growth factor signals also leads to growth signaling autonomy and tumor progression, so control of growth factor expression may prevent tumor progression. This review describes the role of some of the main growth factors whose dysregulation promotes liver tumor progression, and are also key in regenerating the remaining liver following resection. We herein summarize and discuss studies focused on partial hepatectomy and liver carcinogenesis, referring to hepatocyte growth factor, insulin-like growth factor, and epidermal growth factor, as well as their suitability as targets in the treatment of hepatocellular carcinoma. Finally, and given that drugs remain one of the mainstay treatment options in liver carcinogenesis, we have reviewed the current pharmacological approaches approved for clinical use or research targeting these factors.

## 1. Introduction

Liver cancers are the fourth most common cause of cancer-related death worldwide [1]. Their incidence has increased in Western countries over the last two decades, and is expected to continue rising in the future [2]. The most frequently occurring primary malignancy of the liver cancer is hepatocellular carcinoma (HCC), which arises from hepatocytes, and represents at least 80% of primary liver cancers [3,4]. In addition, fibrolamellar HCC, angiosarcoma, lymphoma, and embryonic sarcoma are other liver malignancies of non-cirrhotic origin, but are rare in occurrence [5].

HCC risk factors are parallel to sustained liver injury risk factors, i.e., hepatitis B and C, excessive alcohol intake resulting in alcoholic liver disease, lifestyle patterns, and pathological processes leading to non-alcoholic liver disease, among others [6]. HCC usually develops in patients with underlying cirrhosis (approximately 80%) [7]. From a pathophysiology viewpoint, liver damage produces alterations in the normal hepatic microenvironment and induces inflammation, necrosis, and regeneration. In this scenario, select hepatocyte populations can transform into dysplastic nodules and evolve into liver cancer [8].

Despite recent great improvements in liver cancer therapeutics such as surgery, chemotherapy, and immunotherapy, this disease remains highly lethal due to its aggressive metastases [9,10]. Furthermore, some evidence points to the ability of solid tumors to develop multiple invasion and resistance pathways that allow them to circumvent inhibition by a single signaling pathway [11]. In the specific case of HCC, at the time of diagnosis, tumors may be too large or may have encroached on nearby major blood vessels or metastasized, rendering most HCC patients unsuitable candidates for surgical resection [12,13]. In addition, chemotherapy only leads to moderate improvement of the overall survival of patients, due to its lack of specificity as well as the side effects that are often induced as a result of their significant toxicity. Thus, limitations in HCC treatment result in a poor prognosis, and resistance to traditional radiotherapy and chemotherapies [14]. Unfortunately, patients with advanced HCC have a median overall survival below 1 year [15], and the 5-year recurrence rate of HCC is as high as 70% [16]. The same results (i.e., little effect of chemotherapy, poor prognosis, and treatment resistance) can be seen in other less common types of liver cancers [17]. Therefore, in addition to the mechanisms described so far, there must be others involved in the regulation of liver cancer that are eluding us, and that are playing a very important role in disease progression. This highlights the need to expand our knowledge and understanding of the molecular signaling processes contributing to the poor survival of patients with liver cancer, and that may contribute to the development of new drugs or therapeutics with safer and more effective properties.

In a healthy state, cells in multicellular organisms habitually communicate with one other via growth factors, among other molecules. This communication lets the cells know whether or not to divide [18]. In tumor progression, cell surface receptors can lead to dysregulation resulting in self-sufficiency in growth signaling, one of the major hallmarks of cancer cells. In this sense, growth factor receptors are overexpressed in numerous types of cancer, and may enable the cancer cells to become hyper-responsive to ambient levels of growth factors, and even ligand-independent signaling. Growth factor receptors mediate tumorigenic activity by multiple signaling pathways [19]. During the fetal stage, a large amount of growth factors are produced in the liver e.g., epidermal growth factor (EGF), insulin-like growth factors (IGF-1), and hepatocyte growth factor (HGF), among others. In contrast, many of these are absent or scarce in the normal adult liver [20].

Liver regeneration is a restorative process that follows hepatic parenchymal damage or surgical resection [21]. When liver regeneration is required (i.e., hepatic resection), adult hepatocytes can up-regulate the production of EGF, HGF, and IGF-1, among others [22,23]. This process is dysregulated in the chronically injured liver, and such dysregulation of growth factor production and growth factor receptor signaling in adult hepatocytes plays a crucial role in the development of hepatic cancer [24].

As to the effect of liver regeneration on the malignant hepatic tumor following major surgery, liver regeneration might be driven by the overexpression of EGF, HGF, and IGF-1, which could activate hidden micrometastases and facilitate tumor growth; this results in liver tumor recurrence, and also stimulates intrahepatic tumor propagation due to abnormalities in the cellular signaling pathways [21].

To date, liver resection in cancer patients has evolved significantly, making it a safe surgical option when performed in the appropriate context [25]. Therefore, the complete role of growth factors in carcinogenesis after the surgical procedure (i.e., regeneration, proliferation, outcome after surgery, etc.,) needs to be elucidated. It is also mandatory to decipher the exact role of such factors in response to ischemia-reperfusion (I/R), a procedure commonly performed in liver resection to prevent bleeding and that promotes inflammation and injury, and negatively compromises the regenerative process [26]. In addition, the mechanisms involving endocrine, autocrine, and paracrine signals occurring during liver regeneration can influence dormant micrometastases and tumorigenesis in the remnant liver [27]. Bearing this in mind, achieving a balance between the promotion of post-resection liver regeneration and the inhibition of tumor recurrence has become a major problem in liver surgery. Thus, research advances on the underlying biology and pathophysiology of liver cancer in terms of growth factors and their specific role in carcinogenesis and regeneration, are required to develop effective and novel anticancer therapeutics.

This study reviews the literature focused on growth factor-mediated mechanisms in liver cancer. We have underscored those studies performed in the context of liver resection, both clinical and preclinical studies focusing in particular on HGF, IGF-1, and EGF (Figure 1).

## 2. Growth Factors in Liver Tumorigenesis and Liver Regeneration

After partial resection, liver regeneration is a perfectly regulated and highly coordinated response whose apparent sensor is the need to modify the size of the remaining liver; it involves the action of several growth factors [28,29]. Liver regeneration progression is composed by three main phases: (1) The priming stage, (2) proliferation stage, and (3) termination stage. HGF, EGF, and IGF-1 are involved in the proliferation stage [29].

In addition, the development and progression of liver cancer and liver metastases is a multifactorial process, linked to alterations in some prominent cellular signaling pathways, including dysregulation of HGF, EGF, and IGF-1. These signals result in a self-sufficienct process of growth signaling, and secondarily promote tumor progression [20,29]. Further, after the resection of a liver tumor, non-detected microtumors may likely remain in the non-resected liver, and molecular changes following hepatectomy, including surgical stress responses and I/R injury, can modify tumor growth kinetics [29].

This might explain, at least in part, the high rate of liver cancer recurrence after resection. The latter is a topic of current great interest, frequently approached by authors in recent years. Figure 2 summarizes the HGF, EGF, and IGF-1 molecular pathways, known to play an active role in liver regeneration and tumorigenesis, and to be discussed later in this section.

### 2.1. Hepatocyte Growth Factor (HGF)

The hepatocyte growth factor is crucial to liver recovery after major surgery. The administration of HGF [30,31,32] or an activator of HGF [33] has been associated with an increase in hepatic recovery and of its total body mass, hepatic proliferation, and hepatic function in both healthy and cirrhotic experimental models of liver transplantation (LT) and partial hepatectomy (PH). On the other hand, the role of HGF in the induction of liver tumorigenesis is well known, whereby HGF and its high-affinity receptor, mesenchymal–epithelial transition factor [MET (c-Met)], are closely associated with the onset, progression, and metastastatic process of multiple tumors [34,35]. HGF can induce reactive oxygen species (ROS) signaling, thus increasing cell adhesion, migration, and invasion of liver cancer [36].

The HGF/c-Met axis is involved in cell proliferation, movement, differentiation, invasion, angiogenesis, and apoptosis, by activating multiple downstream signaling pathways [35]. Both HGF and its receptor c-Met are commonly de-regulated in HCC [36]. Zhang et al. showed that the deregulation of HGF/c-Met signaling was implicated in the epithelial–mesenchymal transition (EMT), and resulted in tumor progression [37]. Hence, signaling pathways activated by HGF-c-Met might be useful targets in the prevention of tumor progression [38]. In a study performed in different liver cancer cell lines, activated hepatic stellate cells promoted residual tumor progression by modulation of autophagic survival to proliferation via HGF/c-Met signaling [39]. Furthermore, HGF/c-Met interaction fomented metastasis development from malignant cells in vitro and in vivo [40]. In a study conducted in reduced-size liver transplantation (RSLT), the over-expression of HGF was associated with an up-regulation of cMet expression, resulting in improved liver function [41]. As mentioned above with HGF, the up-regulation of c-Met might be a risk factor in liver surgery conditioning the presence or absence of tumorigeneses. In tumorigenesis, the different pathways activated by HGF are ERK1/2/MAPK and PI3K/Akt [42], leading to the oxidation of heat shock protein 60 and protein disulfide isomerase-induced ERK, and the migration of tumor cells [36,43]. On the other hand, these signaling cascades are known to be the classical pro-survival pathways involved in the reduction of liver injury, and improved liver regeneration after hepatic resections and LT [44]. As in the HGF-cMet axis, the activation of such pro-survival pathways protect against liver injury and promote liver regeneration, but their failure might promote tumor development in the presence of undetectable micrometastases.

In liver regeneration associated with reduced-size orthotopic liver transplantation (RLOT), a direct relationship between HGF and ROS has been established [45]. Therefore, the effects of ROS induced by HGF might be different, depending on the presence or absence of tumorigenesis. It is well known that ROS induces DNA damage, apoptosis, and an inflammatory response. Such effects negatively compromise liver regeneration after ROLT. Thus, the down-regulation of ROS induced by HGF is beneficial in ROLT. However, in the presence of tumorigenesis, HGF induces ROS generation, thus promoting further tumorigenesis [35]. In clinical practice, we should consider that liver grafts with tumorigenesis are discarded from implantation in the recipient. However, undetectable micrometastasis in the small liver graft used in ROLT, and the possibility of cancer recurrence once implanted in the recipients should be not discarded. Bearing this in mind, it is very difficult to elucidate whether we should inhibit or potentiate the effects of HGF, since this growth factor promotes tumorigenesis as well as hepatic regeneration to reestablish the standard liver mass following surgery.

Accordingly, further studies are necessary to elucidate the role of HGF in liver injury and regeneration after hepatic resections and LT. Preclinical studies on HGF evaluating the role of endogenous and exogenous HGF (using pharmacological modulation or transgenic animals) have mainly focused on non-steatotic livers undergoing PH without vascular occlusion. The signaling pathways through which HGF acts, are ERK1/2, decreasing apoptosis and pro-inflammatory mediators such as IL1β [30,31,33,46,47,48]; however, the effects of HGF should be analyzed using the appropriate experimental models of surgery, simulating “real-life” clinical conditions as much as possible. Vascular occlusion, which is usually performed in liver surgery to prevent major bleeding, has not been considered in the experimental designs. In addition, the baseline conditions of the liver (for instance, non-steatotic versus steatotic livers) have not been taken into account in literature reports. It is well known that the baseline status of the liver might also affect the post-operative outcomes, as a result of HGF effects, since the pathological mechanisms of injury and regenerative failure in both types of livers are very different. All these points should be considered in future investigations.

### 2.2. Epidermal Growth Factor (EGF)

Epidermal growth factor is overexpressed during cancer development, acting as a hepatocyte mitogen [49]. Considerable evidence suggests that EGF may be one of the key factors that initiate hepatocyte proliferation, and supports a role for EGF in malignant transformation and tumor progression [49]. This is confirmed by increased EGF expression in liver tumor tissue obtained from patients. Moreover, EGF in the tumor microenvironment may also influence the behavior of cancer cells [50].

Recent literature has addressed the underlying interactions between tumor cells and their microenvironment when studying tumor progression and the metastatic process. EGF promotes DNA synthesis, regeneration, tumor growth, and the progression of tumor cells [51]. In addition, the activation of EGF/EGFR signaling and the overexpression of proteins such as Dickkopf-1 (DKK1) [52], th ehigh-mobility group protein A2 (HMGA2) [53], and PD-L1 [54], or Protein kinase-C (PKC) phosphorylation of p65 [55], are associated with the degree of aggressiveness and the potential to develop intrahepatic metastases. In other cases, EGF cooperates with other cytokines, e.g., TNFα, to promote the motility of tumor cells [56].

Pharmacological approaches aimed to modulate EGF and related pathways have also been tested in the past few years. As an example, in a study conducted in human liver cancer cells, treatment with a dietary component, catecholone, inhibited the EGFR-Akt-ERK signaling pathway [57]. Additionally, the addition of delphinidin (a flavonoid) [58] and ginsenoside Rg3 [59] to the culture medium inhibited the EGFR/Akt/ERK signal axis in liver cancer cells. Da Huang Wan, a traditional herbal medicine, also decreased tumor progression in vitro by modulating EGF [60]. These results indicate that both HGF and EGF might exert their effects via the same signaling pathways i.e., Akt/ERK.

Several authors have pointed out that in major liver surgeries, EGF expression occurs in the early stages after PH, similarly to HGF [48,61]. However, different results have been reported on the role of EGF in liver surgery. Some studies indicate that EGF may not be essential for liver regeneration [62]. In addition, others authors have observed that the administration of AE788, a dual EGF-VEGF-receptor tyrosine kinase inhibitor, can inhibit the auto-phosphorylation of both EGFR and VEGFR, leading to anti-proliferative and anti-angiogenic effects [63,64]. The latter, in association with decreased EGFR expression levels, was associated with no changes in either liver regeneration or hepatic injury [62].

Results from PH have highlighted the beneficial effects of EGF, which has been associated with an increase in liver weight and proliferation markers after its administration [65,66]. In addition, the down-regulation of EGFR was associated with a decrease in hepatocyte proliferation and an increase in cell death [67].

Amphiregulin, another EGFR ligand, has been involved in liver regeneration after RSLT. Exogenous administration of this compound in small-for-size liver grafts decreased liver injury [68]. This was associated with an increase in mTOR, p70S6K, pERK172, and pJNK1/2. Furthermore, in an experimental model of LT obtained from brain-dead donors, EGF was reported to decrease GH (growth hormone) levels in steatotic liver grafts. This was associated with an increase in PCNAs and a reduction in the biochemical and histological parameters of hepatic damage, thus improving the survival rate of recipients [69]. However, in healthy livers, the authors reported the negative effects of GH, reflecting the importance of the type of liver used in a robust study and treatment [69].

Consequently, since EGF promotes tumor progression, and since there are very different study results on the effect of EGF on liver regeneration, intensive preclinical investigations should be conducted on the potential application of EGF in clinical practice since its effects on liver regeneration are dependent on the type of liver surgery (PH versus LT) and the type of the liver (steatotic versus non-steatotic liver); studies on the different drugs or interventions used to either inhibit or activate the effects of EGF are also necessary. In addition, we have to account for the presence of different ligands for EGFR, so the precise role of EGF in liver surgery is difficult to strictly establish. The mechanisms of action of EGF in tumorogenesis should also be considered, as in TNF. In this sense, in hepatic resections and LT, TNF is an inflammatory mediator, and its role in liver regeneration is controversial [70]. Indeed, previous studies suggest that IL-6 and TNF play a crucial role in the first stage of proliferation [71], and other authors have pointed out its negative effects on liver regeneration [72].

### 2.3. Insulin-Like Growth Factor-1 (IGF-1)

IGF-1, a small polypeptide hormone with a similar structure to the insulin protein, is also implicated in liver tumorigenesis by activation of the EMT pathways [73,74].

IGF-1 is a crucial gene in liver cancer, and is highly up-regulated in HCC [75]. Serum IGF-1 levels have been suggested to be predictors of progression and survival in liver cancer patients [74,76]. Recent research efforts have focused on targeting the IGF-1 axis members and the modulation of IGF-1 regulatory pathways in an attempt to find therapeutic options that will inhibit proliferation and invasion by HCC. In this sense, IGF-1 activates PI3K/Akt [77], and induces EMT of HCC via the Stat5 signaling pathway [78], and is also involved in the regulation of the p53 signaling pathway [75].

As with EGF, a different role for IGF-1 has been also described. IGF-1 has been related to proper liver function in LT experimental models [79]. Such benefits might be explained by the increased expression of PI3K/Akt, a decrease in Bim, FasL, apoptosis FoxO3, and the over-expression of SIRT1 and Akt. However, controversial results on the role of IGF-1 have been reported in PH [79,80]. In studies by Desbois-Mouthon et al. [80], the benefits of PH were detected using LIGFREKO—not expressing IGF-1R in hepatocytes and cholangiocytes—whereas the studies by Liu [81] showed no effect of IGF-1 in transgenic mice overexpressing IGFBP-1 (insulin-like growth factor 1 binding protein). These studies were very different (lack of IGF-1R versus IGFBP-1 over-expression) in their analysis of the role of IGF-1, and do not mimic the endogenous or exogenous role of IGF-1 in clinical situations. In addition, a different percentage of resection was used in both studies. In clinical studies, a decrease in its concentration has been correlated with some advanced chronic liver diseases such as cirrhosis [82,83], but levels could be restored after LT [83,84,85]. Furthermore, a higher concentration of IGF-1 has been associated with a favorable outcome after LT (increase in survival, decrease in time spent at the hospital), and a decrease in LD (liver dysfunction) [83]. Of clinical interest, treatment with IGF-1 may also decrease injury to remote organs following liver surgery, even tampering with kidney dysfunctions following LT [79]. As with EGF, the effects of IGF-1 promote tumor progression but could play a role in liver regeneration, considering that IGF-1 (as the other growth factors, HGF and EGF) exerts its action mainly via the same pathways as HGF or EGF; the effects of IGF-1 in promoting tumorigenesis are clear, but its effects on injury and regeneration are controversial. The latter might be explained by the different surgical procedures followed, the types of livers, as well as the types of pharmacological or transgenic modulation of IGF-1.

In summary, targeting EGF, HGF, and IGF-1 seems to be a useful approach to identify an efficient HCC treatment. However, despite all the mentioned potential advantages of targeting these three growth factors to treat HCC, it is necessary to note that, to date, one of the most effective treatments is surgery when possible, especially tumor resections. As a result of this surgical procedure, the liver loses mass and the regenerative pathways are activated.

### 2.4. Hepatocyte Growth Factor, Insulin-Like Growth Factor and Epidermal Growth Factor and Their Link with Vascular Endothelial Growth Factor

Vascular endothelial growth factor (VEGF) expression is up-regulated by the activation of EGF, IGF-1, and HGF receptors [86], reflecting the strong link between VEGF, HGF, IGF-1, and EGF in this review.

In the context of HCC, VEGF acts as in a typical hypervascular tumor. HCC produces and secretes VEGF, thus forming new tumor vessels that provide oxygen and nutrients to cancer cells, thus inducing their growth [87]. As with EGF, IGF-1, and HGF, VEGF also contributes to metastasis development [87]. VEGF causes vascular fragility and decreases cell-to-cell adherence in the endothelium, which might allow cancer cells to migrate into the vascular lumen [88]. In HCC, as in other types of cancer, VEGF shares common pathways with EGF, IGF-1, and HGF, such as MET, EKT, or Akt, among others [89,90,91,92]. Therefore, drugs targeting EGF, IGF-1, and HGF may critically disrupt not only tumorigenesis and progression of HCC but also angiogenesis, and counteract the metastasizing potential of the tumor.

However, there is evidence obtained from PH studies that VEGF levels increase after surgery, and that the infusion of VEGF can decrease injury and increase hepatocyte proliferation [93,94,95]. The latter is required in liver surgery since I/R reduces liver regeneration and is also associated with post-surgical liver failure or graft dysfunction [28]. VEGF protective effects on liver injury and liver regenerative failure have been observed in non-steatotic livers subjected to PH with vascular occlusion. However, VEGF was not beneficial in the presence of steatosis, because it is sequestered by the high circulating sFlt1 levels [94].

There is a notable relationship between I/R, HCC, and the activation of VEGF (involved in both regeneration and tumorigenesis, as mentioned) with EGF, IGF-1, and HGF; this is of great importance, not only because I/R is intrinsic to liver resection but because this process is associated with increased inflammation in the remaining liver tissue. In addition, the effects of these growth factors are dependent on the liver’s baseline status. In consequence, uncontrolled I/R in a damaged liver with possible residual micro-tumors, and with the need for regenerative processes to restore liver size, results in the perfect storm being created, making disease recurrence highly probable. Thus, further investigations on all these factors but considering them as a whole, and an attempt to control them, could become a more powerful therapeutic strategy than trying to regulate them separately. An interesting possibility would be to look for variants of EGF, HGF, IGF-1, and VEGF as has already been done with other molecules; this would foster the regeneration process or at least not hinder it, without affecting tumor progression.

## 3. Pharmacological Approaches Targeting HGF, EGF and IGF-1 Approved for the Treatment of Liver Tumorigenesis

As explained above, HGF, IGF-1, and EGF are key elements in liver regeneration, but the dysfunction of these growth factors also plays an important role in tumorigenesis and cancer recurrence. Thus, the pharmacological blockade proposed to avoid cancer recurrence after or before resection could compromise liver regeneration; the exploration of the clinical effectiveness of drugs against these growth factors in the treatment of liver tumorigenesis is imperative.

### 3.1. Clinical Strategies in Liver Tumorigenesis and Liver Regeneration Based on Growth Factor Modulation

Sorafenib, an oral multikinase inhibitor, was approved as a first-line systemic therapy in advanced HCC [96]. Sorafenib blocks the receptor of tyrosine kinase signaling and inhibits downstream Raf serine/threonine kinase activity, thereby inhibiting the proliferation and survival of tumor cells. However, tumor resistance to sorafenib is the major obstacle to improve HCC patients’ survival. Resistance to sorafenib can be acquired by cancer cells through the activation of pathways via receptor tyrosine kinases [97]. The effect of sorafenib on the interaction between macrophages and hepatocytes was evaluated using polarized THP-1 cells to M1 and M2 macrophages, and a sorafenib-resistant xenograft. This study concluded that M2 macrophages mediated sorafenib resistance by secreting HGF in a feed-forward manner in HCC [96]. Other authors have found elevated HGF expression as an autocrine c-Met activation mechanism in acquired resistance to sorafenib in hepatocellular carcinoma cells [98], and that exosomes derived from HCC cells are responsible for sorafenib resistance in hepatocellular carcinoma in in vivo and in vitro models [93]. Another proposed mechanism suggests that HGF induces sorafenib resistance by inhibiting ERK and STAT3, and subsequently down-regulating Snail and EMT-mesenchymal transition [99].

Other compounds have been evaluated as treatment options for HCC. For instance, in an HCC xenograft model in nude mice, the Chinese herbal formula QHF inhibited metastasis via the HGF/c-Met signaling pathway [100]. Similarly, treatment with resveratrol or deguelin inhibited HCC cell growth and suppressed HGF-induced invasion through down-regulation of the c-Met signaling pathway in vitro [92,101].

Although the landscape of HCC management is changing, Sorafenib remains one of the most effective single-drug therapies [102]. However, tumor resistance to sorafenib and its severe side effects are notorious disadvantages for patients. Several studies point out that the activation of pathways related to the expression of growth factors is of importance in the development of Sorafenib resistance. Thus, current advances in therapies for patients with HCC need to be addressed to better understand these phenomena.

### 3.2. Other Drugs Blocking the HGF/c-Met Axis in Liver Tumorigenesis

Since the HGF/c-Met axis is one of the most understood pathways in HCC, several Met inhibitors have been tested to block the transmission of downstream signals, and find pharmacotherapeutic options to Sorafenib.

Cabozantinib, an inhibitor of multiple receptor tyrosine kinases, has been proposed as second-line therapy in HCC [103]. A double-blind trial performed between 2013 and 2017, that included 707 patients from 19 countries, showed an overall survival of 10.2 months with cabozantinib versus 8.0 months in patients who received placebo. This trial also showed that the median progression-free survival was 5.2 months with cabozantinib and 1.9 months with placebo [104]. Despite the promising results of this c-Met blocker, further research is needed to establish the optimal doses in HCC, since cabozantinib could become the drug of choice in patients previously treated with sorafenib [105].

Tivantinib is another drug that has been recommended for the treatment of patients with HCC [106]. Although its mechanism of action is under discussion [107], the available data suggests that tivantinib acts by selectively blocking the activation of c-Met [108]; accordingly, this drug is proposed to treat patients with MET-high HCC. Kudo and colleagues recently reported a double-blind, placebo-controlled, phase 3 study in the Japanese population, in which they detected a slight increase in the median progression-free survival, with 2.8 months in the tivantinib group versus 2.3 months in the placebo group (*n* = 134 and 61, respectively) [106]. A phase 3, randomized, placebo-controlled study developed between 2012 and 2015 in patients previously treated with sorafenib revealed that patients who received tivantinib (*n* = 226) had an 8.4 month median overall survival versus 9.1 months in the placebo group (*n* = 114) [109]. More research is needed to accept or exclude the use of tivantinib in patients with HCC.

Recently, tepotinib has emerged as an option to treat patients with advanced solid tumors overexpressing MET receptors [110,111]. This drug, a potent and highly selective MET inhibitor, was well tolerated in a phase I trial conducted on 149 patients, and was shown to decrease or stabilize tumor burden [96]. Later, Ryoo and colleagues reported the results of a phase II trial comparing tepotinib versus sorafenib in Asian patients, and showed that tepotinib improved the time to progression compared to sorafenib (2.9 months versus 1.4 months, respectively) [112].

### 3.3. Pharmacological Approaches to Modulate EGF and IGF-1 Pathways in Liver Tumorigenesis

As previously mentioned, overexpression of the epidermal growth factor receptor (EGFR) is frequent in HCC. EGFR activation is a potential determinant of primary resistance of HCC cells to sorafenib [113]. For instance, KIAA1199 protein induced tolerance to sorafenib and metastases development by activating the EGF/EGFR-dependent EMT-mesenchymal transition program [114].

Gefitinib, a reversible inhibitor of EGFR-TK, was approved in 2003 for the treatment of patients with advanced non-small-cell lung cancer [112]. According to Höpfner and colleagues, gefitinib potently suppresses the growth of several tumors, including HCC, by inducing apoptosis via suppression of antiapoptotic Bcl-2 and Bcl-XL expression, in addition to cell cycle arrest [112]. Nevertheless, pre-clinical models showed that gefitinib decreases chemical and hormonal HCC but, interestingly, no prevention response is observed [112]; in addition, several mutations reported on the tyrosine-kinase domain have restricted its use in HCC patients [115].

Erlotinib is part of the first generation of EGRF inhibitors, and was approved by the FDA in 2004 for the treatment of solid tumors [115]. Studies developed in human HCC cells Huh-7 showed that erlotinib’s mechanism of action, at least in part, results from the inhibition of mitogen-activated protein (MAP)-kinase and STAT pathways, leading to down-regulation of antiapoptotic factors Bcl-2 and Bcl-XL, inducing cell cycle arrest and apoptosis [116,117]. According to the systematic review by Zhang and colleagues published in 2016, the clinical use of erlotinib may be a relatively efficacious and well-tolerated treatment for advanced HCC [118].

Lapatinib is also a reversible inhibitor of EGFR tyrosine kinase activity [119,120] and its mechanism of action in a xenograft model was found to be the induction of autophagic cell death leading to the inhibition of HCC tumor growth [121]. The phase II study on the efficacy of lapatinib revealed a median progression-free survival of 1.9 months, and a median overall survival of 12.6 months in patients with advanced HCC [122]. Similar results in the median progression-free survival were reported by Ramanathan and colleagues [123].

Afatinib is an irreversible EGFR inhibitor used to treat patients with relapsed or refractory solid tumors, and approved by the FDA in 2013 [115]. Studies conducted with the HCC cell line Huh-7 revealed that, at least in part, afatinib exerts its action by regulating the ERK-VEGF/MMP9 signaling pathway, and decreasing the epithelial–mesenchymal transition and tumorigenesis [124].

On the other hand, not many pharmacological options based on the inhibition of the IGF-1 receptor have been tested to manage HCC. Esculetin (6,7-dihydroxy coumarin), a coumarin derivative extracted from natural plants, has been reported to have anticancer activity. In vitro, the administration of esculetin inactivated IGF-1, and inhibited the PI3K/Akt pathways [75]. Additionally, nicotinamide had a protective effect in HCC by inhibiting IGF-1 and balancing the nuclear factor erythroid 2–related factor 2 (Nrf2) and the PKB nicotinamide ratio, both in in vivo and in vitro HCC models [125]. Finally, linsitinib and brigatinib, although still not tested as treatment of HCC, have shown promising results in non-small cell lung cancer [126,127,128,129], providing a possible use for them as an alternative/complementary drug to sorafenib.

Under these conditions, using drugs against EGF, HGF, and IGF-1 as complementary therapy to prevent the progression of any residual tumor cells may be very counterproductive because these three growth factors are also actively involved in the regenerative process and, along with other molecules, are the main mediators of their coordination (see Table 1). Nevertheless, until the roles of HGF, IGF-1, and EGF are better understood, the current pharmacological approach will continue to be used in clinical practice.

### 3.4. Role of HGF, EGF and IGF-1 in Liver Hepatectomy: Implications of External Regulation

After resection of a liver tumor, non-detected microtumors may remain in the non-resected liver. This might explain, at least in part, the high rate of recurrence of HCC after resection. It is clear that the use of tumor growth blockers in a regenerating liver with possible microtumors might alleviate tumor progression but with deleterious effects on the restoration of the remaining hepatic tissue. Likewise, the administration of activators or recombinants of the growth factors mentioned above, as a strategy to improve post-surgical regeneration, would not be appropriate since tumor progression would be promoted.

Despite the potential key role of EGF, HGF, and IGF-1 in PH and LT, information on treatment with growth factors in the clinic is scarce. To date, robust knowledge on the role of these growth factors in liver regeneration has been obtained in preclinical models that have shown that HGF may have regenerative effects [30,130,131]). The administration of HGF [30,31,32] or an activator of HGF [33] has been associated with an increase in the recovery of the liver and its total body mass, hepatic proliferation, and hepatic function in both healthy and cirrhotic experimental models of liver transplantation (LT) and PH. This has also been detected in the livers of LT and PH patients [132,133].

EGF is another growth factor often studied in major liver surgeries. Some experimental articles point out that the expression of HGF occurs in the early stages after PH [48,61]. However, different results have been reported in the literature on the role of EGF in liver surgery, suggesting that EGF may not be essential for liver regeneration [62]. On the other hand, other studies in PH highlight the beneficial effects of EGF that has been associated with an increase in liver weight and proliferation markers after its administration [65,66]. In addition, the down-regulation of EGFR was associated with a decrease in hepatocyte proliferation and an increase in cell death [67].

As with other growth factors, a different role of IGF-1 has been described. IGF-1 has been associated with appropriate liver function in experimental models of LT [79] or PH [80], although it might not be essential in PH [81]. Greater IGF-1 concentrations have been associated with a favorable outcome after LT (increased survival, decrease in time spent at the hospital), and a decrease in LD (liver dysfunction) [83].

There are not many studies on the role of the different growth factors (HGF, EGF, and IGF-1) in patients who undergo liver surgeries in the context of tumorigenesis. The scenario that can be presented in the medical clinic is that after PH, undetected microtumors remain in healthy tissue, and a reasonable approach after a tentative recombinant growth factors treatment could result as counterproductive since healthy tissue could undergo regeneration, but the carcinogenic effect of these same pathways might activate microtumor growth, and result in cancer recurrence. A similar situation could occur in LT with undetected microtumors followed by exogenous growth factor treatment. Further studies are required to elucidate the exact role of growth factors in the context of PH and HCC (including the presence of vascular occlusion) since the existing data are scarce, controversial, or lacking solid evidence as reflected in the recent systematic review by Hoffmann et al. [23].

Pharmacological blockade of EGF, HGF, and IGF-1 could play a key role in the prevention of cancer recurrence fostered by undetected microtumors after PH. However, the use of blockers could be detrimental because, as previously discussed, these pathways are active in liver regeneration. The use of Sorafenib after liver resection in intermediate-stage and advanced HCC has been shown to increase overall survival in the surgery + sorafenib group, when compared with the only-surgery group (18.6 vs. 11.9 months, respectively), but there was no differecence in the median time to recurrence [134]. In addition, the use of Sorafenib after PH was associated with lower tumor recurrence compared with the surgery group (44.1% vs. 75%, respectively), with 12 months of median disease-free survival in the Sorafenib + surgery group, and 10 months in the only-surgery group [135]. However, another report referenced no difference in overall survival between the Soranefib + surgery group compared with the surgery group, although the use of sorafenib after surgery prolonged the disease-free survival (5.2 months vs. 1.8 months, respectively) [136]. Similar results were reported by Huang and colleagues, which led them to propose sorafenib therapy after curative hepatectomy in HCC [137]. Conversely, two meta-analyses showed no significant benefit of the administration of sorafenib after surgery [138,139]

These conflicting reports could be due to the heterogeneous expression of growth factor receptors in HCC. A study reported by Lee and colleagues in 2013 showed that overexpression of the c-MET receptor is present in 27.9% [140] and up to 70% of patients with HCC [141].

Conversely, after unsuccessful pharmacological treatment against HGF, EGF, and IGF-1, it is common to opt for surgery in order to prevent further tumor growth [142]. In this situation, the risk is endangering the regeneration process due to the potential residual amount of drugs, a problem that could be excluded if surgery takes place once the drug is eliminated, which depends entirely on the pharmacokinetics of each molecule. To our knowledge, there is little evidence recommending the use of drugs against growth factors before PH, except for a case report published by Barbier and colleagues, where they presented Downstaging HCC induced by treatment with Sorafenib before surgery, and with no associated regeneration problems [143]. Similar results were presented in another case report in which sorafenib was administered before liver resection, suggesting that sorafenib has the potential to downstage advanced HCC [144].

Despite the availability of several drugs that can block HGF, IGF-1, and EGF (such as Sorafenib, Cabozantinib, Gefitinib, Lapatinib, among others previously described), most have been used to prevent tumor growth, but once surgery is the next therapeutic strategy, these drugs should be withdrawn, although no robust guidelines have been established. Likewise, understanding the effect of these blockers on regeneration and/or cancer recurrence before surgery is pivotal. To our knowledge, Sorafenib is the only drug that has been tested before and after liver resection, and has shown promising results in terms of regeneration and cancer recurrence (Figure 3). However, further studies using Sorafenib and other drugs must be conducted in order to propose new therapeutic approaches associating drugs and surgery.

The data suggest (i) blocking growth factor signaling after or before liver resection appears to not modify the regeneration process; (ii) another regeneration pathway(s) could be activated in these conditions; (iii) given the controversial reports, we cannot a suggest receptor blockade as prophylactic treatment after PH to prevent cancer recurrence mediated by undetected microtumors; and (iv) more clinical studies are needed to understand the role of growth factor signaling in liver regeneration and cancer recurrence, since the available data remains highly limited and controversial, thus precluding any accurate conclusions to date.

## 4. Concluding Remarks

Hepatocellular carcinoma is the second most common cause of cancer-related deaths worldwide and has a high propensity to metastasize. Hepatic resection and liver transplantation have been the mainstay curative treatments in liver cancer patients although many suffer early recurrence within 1 year after liver resection and die. In addition, the process of liver regeneration after hepatectomy can promote tumor growth and activate occult microfocal lesions leading to tumor recurrence.

After hepatic resection, adult hepatocytes can up-regulate the production of particular growth factors such as EGF, HGF, and IGF-1, among others. Their activity is essential to the recovery of liver function after major surgery, such as PH or LT. Dysregulation of EGF, HGF, and IGF-1 plays a crucial role in the development of hepatic cancer and metastases.

Reported data obtained from EGF, HGF, and IGF-1 studies on the different aspects of hepatocarcinogenesis are mostly focused on deciphering the molecular mechanisms, and not much information has been reported in terms of their administration in association with surgical procedures. Further, a large number of growth factors are expressed in both HCC and in the regenerating liver, but their function and potential synergies have not been completely characterized.

Since EGF, HFG, and IGF-1 induce specific growth signals to stimulate cell proliferation, they should also be considered as a possibility to counteract I/R injury given their physiological regenerative role in liver injuries. Although cancer patients undergoing hepatic resection are very common, there are no studies in cancer patients subjected to liver resections that evaluate mass recovery following surgery. This is a great disadvantage because in the clinical scenario, after resection, other surgical and clinical parameters are very relevant, such as vascular occlusion, the extension of the resected liver, elderly patients, hepatic steatosis, obesity, etc. The differential role of HGF, EGF, IGF-1, and VEGF depend on the surgical procedure and the type of the liver, thus confirming the urgent need for further studies in appropriate experimental models, and with livers that are as similar as possible to those undergoing surgical procedures, as well as selecting drugs that specifically regulate each growth factor.

After liver tumor resection, non-detected microtumors most likely remain in the non-resected liver, which might result in a high rate of recurrence of HCC. Thus, the use of tumor growth blockers in a regenerating liver with microtumors might alleviate tumor progression but with associated deleterious effects on the restoration of the remaining hepatic tissue. To solve this problem, variants of EGF, HGF, IGF-1, and VEGFA that promote liver regeneration while preventing detrimental effects on tumor progression should be investigated. We must bear in mind that the use of drugs could be counterproductive after liver resection, and it is necessary to fully understand the molecular pathways underlying the role of HGF, EGF, and IGF-1 in liver regeneration and tumorigenesis. In addition, further research is needed to adapt the use of existing drugs after PH or to develop new ones with a different approach, perhaps by regulating signals upstream of receptor activation. To date, the limited evidence on HGF, EGF, and IGF-1 and their effect on liver regeneration seems to support a beneficial effect on the regeneration process or in cancer recurrence. Undoubtedly, further evidence obtained from clinical models is required befor suggesting the administration of growth factors to promote liver regeneration after PH, without the risk of cancer recurrence, nor the use of pharmacological blockers to inhibit cancer recurrence after PH, without the risk of compromising liver regeneration.

In conclusion, a better understanding of the relationship between liver regeneration and liver tumor propagation is essential for patients suffering from primary and secondary liver tumors. Future studies to identify the dynamics of the regenerating liver and liver carcinogenesis in terms of overexpression, dysfunction, or interactions of these growth factors and their receptors are mandatory to identify targets that will allow the development of effective approaches in the treatment of HCC, and improve regeneration after tumor resection.


## Figures and Tables

**Figure 1 biomedicines-09-01158-f001:**
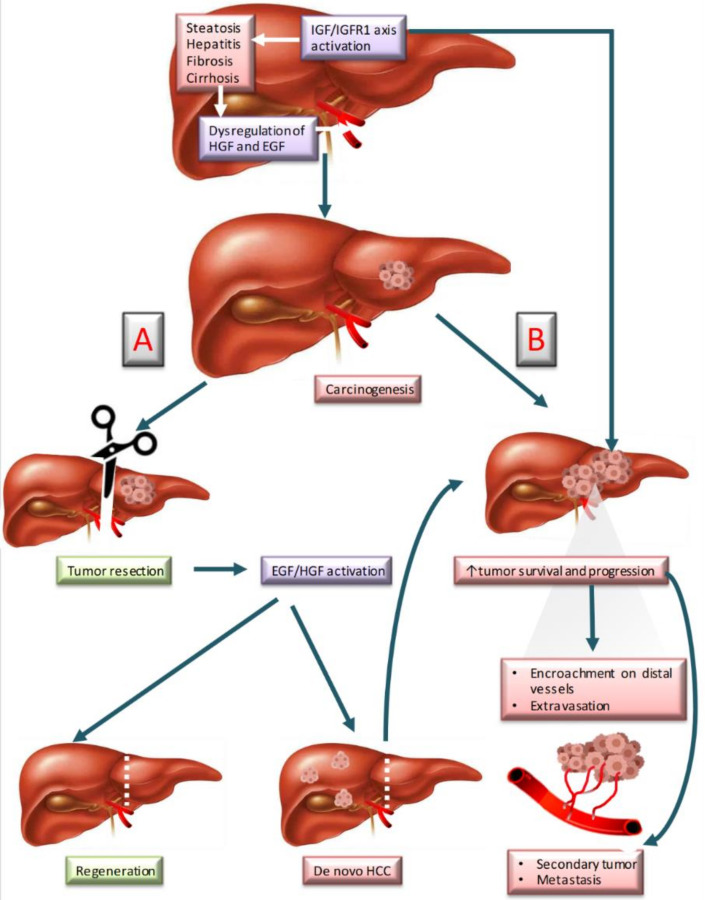
Liver carcinogenesis and regeneration involving HGF, IGF-1, and EGF. (**A**) Normal regeneration after resection. (**B**) Cancer recurrence after resection. Abbreviations: EGF; epidermal growth factor; HCC, hepatocarcinoma; HGF, Hepatocyte growth factor; IGF-1, Insulin-like growth factor-1; IGFR-1, Insulin-like growth factor receptor-1.

**Figure 2 biomedicines-09-01158-f002:**
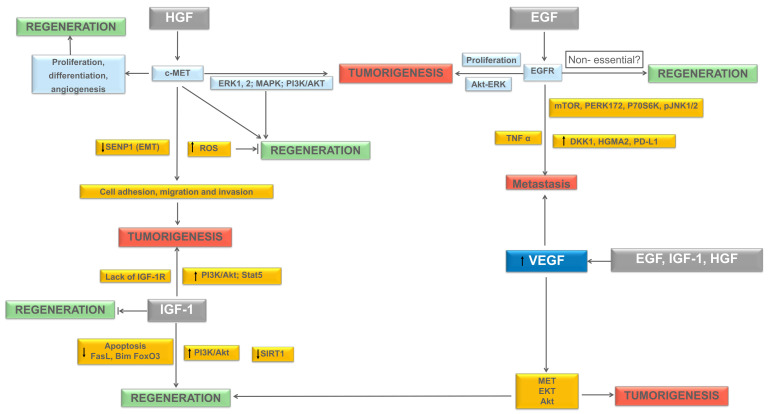
Molecular pathways underlying liver regeneration and tumorigenesis. Proposed pathways of HGF, EGF, IGF-1, and their link to VEGF in inducing regeneration and/or tumorigenesis. Abbreviations: Akt, Protein kinase B; c-Met, Mesenchymal–epithelial transition factor; DKK1, Dickkopf-1; ECM, Extracellular matrix; EGF, Epidermal growth factor; EGFR, Epidermal growth factor receptor; ERK, Extracellular signal-regulated kinase; FasL, Fas Ligand; FoxO3a, Forkhead box protein O3a; HGF, Hepatocyte growth factor; HMGA2, High-mobility group protein A2; IGF-1, Insulin-like growth factor 1; IGF-1 R, Insulin-like growth factor receptor; MAPK, Mitogen-activated protein kinase; MET, Mesenchymal–epithelial transition factor; mTOR, mammalian target of rapamycin; PERK, protein kinase R-like endoplasmic reticulum kinase; PI3K, Phosphoinositide 3-kinase; PD-L1, Programmed death-ligand 1; ROS, Reactive oxygen species. SIRT1, Deacetylase sirtuin1; STAT, Signal transducer and activator of transcription; TNFα, Tumor necrosis factor-alpha; VEGF, Vascular endothelial growth factor. →: induce the expression; ⇥: blockadge of expression; ↑: increment; ↓: decrease.

**Figure 3 biomedicines-09-01158-f003:**
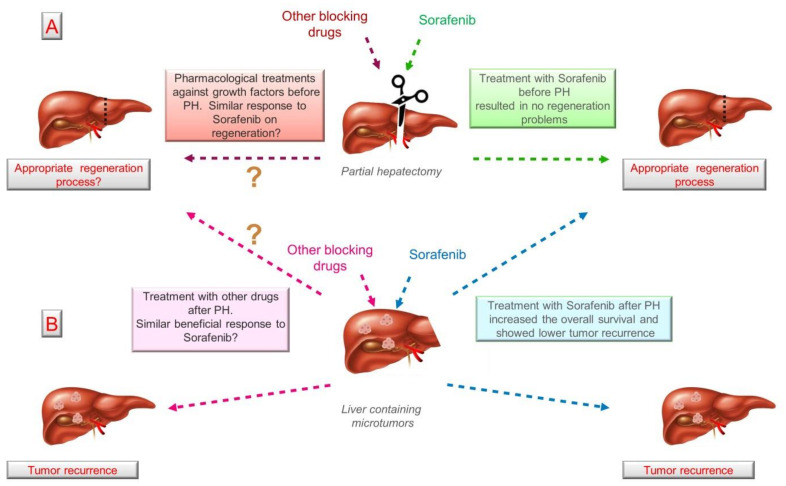
Pharmacological regulation of growth factors and its effect on regeneration and cancer recurrence. (**A**) According to the limited number of studies, treatment with Sorafenib seems does not to affect liver regeneration and could provide benefits to patients after liver resection. (**B**) No other drugs targeting HGF, EGF, and IGF-1 have been tested with similar objectives. ? means unknown. Abbreviations: EGF; epidermal growth factor; HGF, Hepatocyte growth factor; IGF-1, Insulin-like growth factor 1; PH, partial hepatectomy.

**Table 1 biomedicines-09-01158-t001:** Overview of the pharmacological approach with HGF, EGF, and IGF-1 in liver regeneration and tumorigenesis.

Growth Factor	Essential to Liver Regeneration	Inductor of Tumorigenesis	Mechanism (s) Reported Inducing Tumorigenesis	Drug(s) Used to Block the Receptor (Approved or Under Testing in Clinical Trials)	Mechanism of Action
HGF	Yes [30,31,32]	Yes, overexpression of c-MET [35]	ROS, migration and invasion of tumor [36]	Sorafenib Cabozantinib Tivantinib Tepotinib	Multikinase inhibitor [96] Inhibitor of multiple receptor tyrosine kinases [103] A selective blocker of c-Met? [108] High selective MET inhibitor [110]
EGF	Yes [65,66]	Yes, overexpression of EGFR [49]	Activation of EGF-EGFR signaling, overexpressionn of DKK1, HMGA2 [53]; PD-L1 [54]; Phosphorylation of p65 [55]	Gefitinib Erlotinib Lapatinib Afatinib	Reversible inhibitor of EGFR-TK [115] EGRF inhibitor [115] Reversible inhibitor of EGFR [119] Irreversible EGFR inhibitor [124]
IGF-1	Yes [79,80]	Yes, up-regulation of IGF-1 gene [75]	Activation of EMT pathways [74,75,76]	Nicotinamide Linsitinib and Brigatinib	Inhibitor of IGF-1 [125] Inhibitors of IGF-1 [126,127,128,129]

Abbreviations: c-MET, mesenchymal–epithelial transition factor; DKK1, Dickkopf-1EGF; epidermal growth factor; EGFR, Epidermal growth factor receptor; EMT, Epithelial-mesenchymal transition; HCC, hepatocarcinoma; HGF, Hepatocyte growth factor; HMGA2, High-mobility group protein A2; IGF-1, Insulin-like growth factor; PD-L1, Programmed death-ligand 1; ROS, Reactive oxygen species.

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
