# Peer review of "Insights into Growth Factors in Liver Carcinogenesis and Regeneration: An Ongoing Debate on Minimizing Cancer Recurrence after Liver Resection"

_biomedicines, 2021, doi:10.3390/biomedicines9091158_

Round 1
Reviewer 1 Report
In the manuscript entitled “Insights into growth factors in liver carcinogenesis and regeneration: an ongoing debate about minimizing cancer recurrence after liver resection” Ana I. Álvarez-Mercado et al. reviewed and discussed studies in the setting of partial hepatectomy and liver carcinogenesis involving hepatocyte growth factor, insulin-like growth factor and epidermal growth factor as well as their suitability as targets for the treatment of hepatocellular carcinoma.
Basically, the topic of this review is of great interest is of potential interest, nevertheless, the manuscript needs an extended revision
Nevertheless, the paper needs minor comments:
Please check the references in the first part of the introduction. The first three references are not inherent to the argument discussed. The same comment is valid also for reference 7. For reference 9, to look for a more representative review in that field is highly reccomended.
Please check for English revision:
- Lines 139 – 142
- Line 178
- Lines 217 - 224
- Line 285
Author Response
Dear reviewer,
First of all, we appreciate very much your consideration on our manuscript entitled “Insights into growth factors in liver carcinogenesis and regeneration: an ongoing debate about minimizing cancer recurrence after liver resection".
Considering your suggestion, we enclose the revised version of our manuscript, in which it has been modified in accordance with your request.
In addition, the manuscript have been edited and revised by a Native English Expert This paper has been reviewed and edited by Deborah D. Aleman-Hoey, M.D., Ph.D., a Native English Expert in Medical translations from Spanish to English with 43 years of experience.
Reviewer 2 Report
Comments on Review "Insights into growth factors in liver carcinogenesis and regeneration: an ongoing debate about minimizing cancer recurrence after liver resection". I would like to congratulate the authors with a very extensive review on the role of growth factors in liver regeneration and HCC proliferation. Pharmacological approaches targeting HGF, EGF and IGF-1 are also nicely decribed and discussed. However I have a few comments. This is a high volume manuscript, unfortunately containing a lot of repetitions, duplications etc. The english language also needs attention, as sometimes it is not easy to understand the idea of the sentence. Taking into account the volume of the manuscript I would suggest splitting it in two parts: Role of the growth factors and Pharmacological approaches targeting these growth factors. Easier editing and easier reading is sometimes rewarding.Author Response
First of all, we appreciate very much your consideration on our manuscript entitled “Insights into growth factors in liver carcinogenesis and regeneration: an ongoing debate about minimizing cancer recurrence after liver resection".
Dear reviewer,
Considering your suggestion, we enclose the revised version of our manuscript, in which it has been modified in accordance with your suggestions.
In addition, the manuscript have been edited and revised by a Native English Expert. This paper has been reviewed and edited by Deborah D. Aleman-Hoey, M.D., Ph.D., a Native English Expert in Medical translations from Spanish to English with 43 years of experience.
This manuscript is a resubmission of an earlier submission. The following is a list of the peer review reports and author responses from that submission.
Round 1
Reviewer 1 Report
In this review, the authors discussed in detail the role of growth factors in liver cancer. The authors wrote less detailed and difficult to follow review pertaining to this topic. However there are few concerns to be handled. One of our major concerns is that in 2020, a systematic review published by Hoffman et al. discussing a similar theme was published (Hoffmann, K., Nagel, A.J., Tanabe, K. et al. Markers of liver regeneration—the role of growth factors and cytokines: a systematic review. BMC Surg 20, 31 (2020). https://doi.org/10.1186/s12893-019-0664-8). Hence, we are concerned with the loss of novelty in this particular topic. Few other minor concerns are related to the writing flow of some sections like in line 127 and 168 where the authors need to rearrange the lines so as to keep the focus of the discussion to the growth factors instead of miRNAs. While other things that need to be addressed is the clarity of some sentences, such as in line 427, or some typos (table 1, column 3 title) and grammatical errors (line 105).
Reviewer 2 Report
The authors present a review on an ongoing debate, whether liver regneration itself promotes growth of tumor cells and carcinogenesis.
The manuscript separately lists different factors which are involved in carcinogenesis and liver regneration. Still, the manuscript does not provide a lot of insight, how liver regneration influences tumor cells.
Further the focus on HCC makes the interpretation of results even more difficult. It clearly is a difference, whether it is a de novo HCC in a cirrhotic liver or a growth of a micrometastases in a non cirrhotic liver. Additionally, liver regneration in cirrhotic patients is impaired.
Even if this would need conceptual rewriting of the whole mansucript, my suggestion would be to focus on studies investigating tumor growth (not restricted to HCC) in the regenerating liver, rather than reporting studies which found well known growth factors.